# Observability Study on Passive Target Localization by Conic–Angle Measurements

**DOI:** 10.3390/s21196439

**Published:** 2021-09-27

**Authors:** Taeil Suh, Woochan Kim

**Affiliations:** Naval Combat Systems PMO, Maritime Technology Research Institute, Agency Defense Development, Changwon-si 51698, Korea; hagnu@add.re.kr

**Keywords:** observability analysis, target tracking, passive target localization, passive linear array sensor

## Abstract

Information from a passive linear array sensor is related to the conic angle formed by a target and the sensor in three-dimensional (3D) space so that the target localization system using the sensor should be also designed in 3D space. This paper presents an observability study of a passive target localization system created using conic angle information. The study includes the analysis of the sensor maneuver requirement needed to achieve system observability and simulations to demonstrate the results of the analytic scheme. The proposed sensor maneuver requirements satisfy the system observability conditions by using the local linearization approach of the Fisher information matrix. It is also shown that this requirement can be mitigated for special cases in which the depth difference between the sensor and the target is given. Using the simulation, it is shown that sensors following the proposed scheme are able to obtain meaningful information that can be used to estimate 3D target states.

## 1. Introduction

As passive sensors can detect a target without transmitting signals, they are widely used in surveillance areas where the sensor needs to be concealed. In general, observations from these sensors demonstrate no distance information (measurement), and the relative target distance from the sensor needs to be estimated using the observed data for localization. 

The target state estimation technique using these sensors is commonly referred to as target motion analysis (TMA). For two-dimensional (2D) TMA, bearings-only TMA (BoTMA) is a representative type of the 2D TMA, which assumes that the target and the sensor exist on the same plane used for estimating target states using in-plane azimuth information. For conventional three-dimensional (3D) TMA systems, bearing and elevation information are used to estimate the 3D target position as well as the velocity.

In this paper, a passive linear array sensor is considered. From now on, “passive linear array sensor” will be referred to as “sensor” for short. Information from the sensor is related to a conic angle, which is formed by the relative states of the target and the sensor in 3D space due to the beam characteristics of the sensor. The information is neither azimuth nor elevation, so it is not applicable to conventional 2D/3D TMA algorithms. In addition, the target localization system using the conic angle information should be designed in 3D space. We define this system as conic angles-only TMA (CoTMA) and analyze its system observability in this paper. 

Most previous TMA-related studies have focused on proposing estimation algorithms to overcome system nonlinearity or on analyzing system observability to prove the possibility of estimating target states using the given information.

The following estimation algorithms for conventional TMA were previously proposed: 

Maximum likelihood (ML) approaches estimate the nonrandom parameters of the target states that maximize the measurement likelihood with respect to a set of measurements [1]. The ML for nonlinear measurements was proposed in [2], and the ML for pseudo-linearized measurements was proposed in [3]. 

For random parameters, Bayesian approach estimators are used. Minimum mean square error (MMSE) is one of the Bayesian estimation methods, and the Kalman Filter is a representative MMSE algorithm. The extended Kalman filter (EKF) uses a Taylor extension and performs forced linearization to maintain the Gaussian probability density function (pdf) of the target estimates [1]. The unscented Kalman filter and the cubature Kalman filter use set of sigma points that can represent the higher order terms of the Taylor extension to reduce the linearization error [4,5]. Pseudo-measurement filter and modified gain EKF utilize pseudo-linearization using modified measurement equations to linearize the system [6,7]. The range-parameterized EKF improve the problem of the pdf estimates converging to the local MMSE by constructing a bank of EKFs with different distances [8]. An approximation method of the nonlinear measurement likelihood using a mixture of linear Gaussian components was proposed in [9].

Particle filters are sub-optimal estimators for nonlinear systems [10]. They consist of a set of particles to represent the nonlinear and nonGaussian pdf without approximated linearization. The auxiliary particle filter, the regularized particle filter, and the local linearized particle filter methods are variants of the particle filters and were given in [11,12,13], respectively.

Data association algorithms for bearings-only tracking in cluttered environments were proposed in [14,15,16,17].

System observability analysis, which determines whether the parameters being estimated can be inferred from a given observation, should precede the adoption of a TMA algorithm. Until system observability is achieved, no TMA algorithm can guarantee the convergence of its estimates.

In [18], the BoTMA observability for nonmaneuvering targets was analyzed. Additionally, its extended study of 3D TMA was derived from [19]. The observability conditions for target maneuvers with dynamic models of the *N*th-order were given in [20]. A linearized structure of the pseudo-measurements was used to analyze the observability of passive homing guidance applications [21]. In [22], an observability study of a range-only sensor was produced. A comparison of the observability of the range-only and bearings-only approaches was shown in [23,24]. Trajectory optimization methods associated with observability were proposed in [25,26,27], and work regarding matrix optimization for sensor networks can be seen in [28,29,30]. At present, no observability analysis has been conducted for the CoTMA system. 

In this paper, our CoTMA system observability study includes: Designing the system model;Deriving the observability requirements needed to achieve an observable system;Proposing the sensor maneuver strategy to satisfy those requirements;Analyzing a special CoTMA case that can mitigate the requirements;Demonstrating the results of the analytic scheme using simulations.

As previously mentioned, a CoTMA system is designed to estimate the position and velocity of a target in 3D space. System modeling includes the design of a target dynamic equation and the design of a conic angle measurement equation for the sensor.

In this paper, the system observability of the CoTMA is derived using the approach from [22]. This approach uses a Fisher information matrix (FIM) as an observability index and analyzes the nonsigularity condition of the FIM. This condition relates to the relative state of the sensor and the target, and a sensor maneuver strategy is proposed to satisfy the system observability requirements.

Furthermore, a special CoTMA case is also considered. In the anti-ship submarine warfare, the relative depth of the submarine (operator’s own ship) from the surface combatant (target) is equal to the diving depth of the submarine, which is determined by the navigation instrumentation of the ship. This information provides a significant advantage in establishing system observability, and this special CoTMA case is defined as conic angle with depth-supported TMA (CDTMA) in this paper.

The observability studies for the CoTMA and the CDTMA are demonstrated through numerical experiments.

The rest of this paper is organized as follows: Section 2 presents the CoTMA system modeling. As the motivation of this paper, the influence of the vertical incidence angle for the passive linear array sensor is shown in Section 3. The observability analysis of the CoTMA system is presented in Section 4. Section 5 presents the system observability analysis for the CDTMA. The numerical experiments are shown in Section 6. Concluding remarks are given in Section 7.

## 2. System Modeling

Figure 1 shows the geometric relationship between the target and the sensor. The state vectors of the target and the sensor consist of position and velocity in 3D space, and can be calculated as
(1)$$x_{k}^{\tau ,i}\equiv {\left[\begin{array}{cc} {\left(p_{k}^{\tau ,i}\right)}^{T} & {\left(v_{k}^{\tau ,i}\right)}^{T} \end{array}\right]}^{T},\tau \in t,s,$$
(2)$$p_{k}^{\tau ,i}={\left[\begin{array}{c} x_{k}^{\tau ,i}\\ y_{k}^{\tau ,i}\\ z_{k}^{\tau ,i} \end{array}\right]}^{T},\;v_{k}^{\tau ,i}={\left[\begin{array}{c} {\dot{x}}_{k}^{\tau ,i}\\ {\dot{y}}_{k}^{\tau ,i}\\ {\dot{z}}_{k}^{\tau ,i} \end{array}\right]}^{T},\;$$
where p, v, and k denote the position vector, the velocity vector, and the time index, respectively. τ is the object indicator that τ=t is the target of, and τ=s is the sensor. The superscript of the vectors i means that the vectors are represented in an inertial coordinate system.

**Figure 1 sensors-21-06439-f001:**
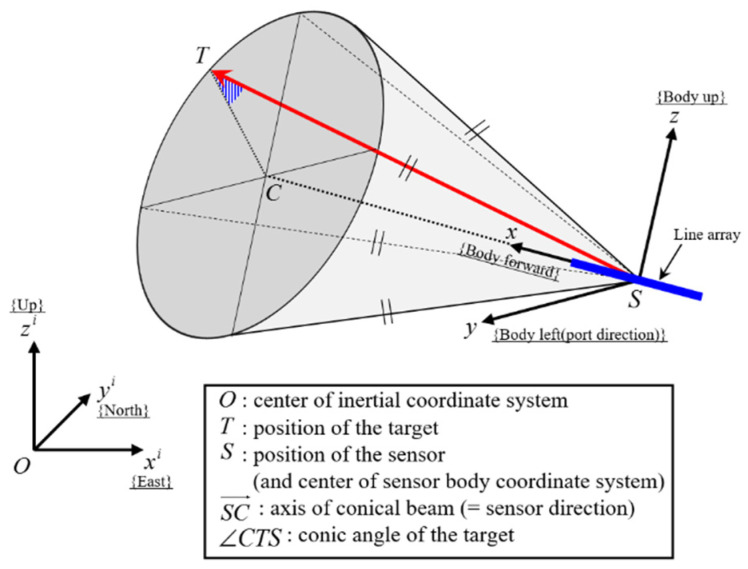
Relative geometry of the target sensor in 3D space.

### 2.1. Target Dynamics

Given the target states at *k − 1* ($x_{k-1}^{t,i}$), a target state at *k* ($x_{k}^{t,i}$) is propagated by a linear dynamics as
(3)$$x_{k}^{t,i}=F\left(t_{k-1,k}\right)x_{k-1}^{t,i},$$
where $t_{\ell ,m}$ denotes a transition time interval from ℓ to m, and F is the state transition matrix. Here, if we assume that the target follows constant velocity motion, then the F is
(4)$$F\left(t_{k-1,k}\right)=\left[\begin{array}{cc} I_{3} & t_{k-1,k}I_{3}\\ O_{3} & I_{3} \end{array}\right],$$
where $I_{n}$ and $O_{n}$ denote the n×n identity matrix and the n×n zero matrix, respectively.

### 2.2. Sensor

As shown in Figure 1, the conic angle is determined by the relative position vector of the target sensor, which is represented in the (sensor-centered) sensor body coordinate system. Coordinate transformation is required to formulate the conic angle in the body coordinates with the relative position in the inertial coordinates. One can define a rotation matrix from the inertial coordinates to the body coordinates at k by
(5)$$C_{k}^{i\to b}=C_{rx}^{}C_{ry}^{}C_{rz}^{}$$
where
(6)$$C_{rx}^{}=\left[\begin{array}{ccc} 1 & 0 & 0\\ 0 & \cos\phi_{k} & \sin\phi_{k}\\ 0 & -\sin\phi_{k} & \cos\phi_{k} \end{array}\right],$$
(7)$$C_{ry}^{}=\left[\begin{array}{ccc} \cos\varphi_{k} & 0 & -\sin\varphi_{k}\\ 0 & 1 & 0\\ \sin\varphi_{k} & 0 & \cos\varphi_{k} \end{array}\right],$$
(8)$$C_{rz}^{}=\left[\begin{array}{ccc} \cos\psi_{k} & \sin\psi_{k} & 0\\ -\sin\psi_{k} & \cos\psi_{k} & 0\\ 0 & 0 & 1 \end{array}\right],$$
and where $\phi_{k}$, $\varphi_{k}$, and $\psi_{k}$ are the Euler angles, which indicate the direction of the sensor with respect to the inertial coordinates. Using Equation (5), the relative target position $p_{k}$ of the body coordinates is calculated as
(9)$$p_{k}^{}=C_{k}^{i\to b}H\left(x_{k}^{t,i}-x_{k}^{s,i}\right)\equiv {\left[\begin{array}{ccc} x_{k} & y_{k} & z_{k} \end{array}\right]}^{T},$$
where $x_{k}^{s,i}$ is the sensor state at time k and is assumed to be known. H is a linear projection matrix, which can be represented as
(10)$$H=\left[\begin{array}{cc} I_{3} & O_{3} \end{array}\right].$$

Note that the origin of the body coordinates is located at the sensor position $p_{k}^{s,i}=Hx_{k}^{s,i}$ in the inertial coordinates.

The measurement equation consists of a nonlinear function of the relative target position $p_{k}$ and the measurement error $\nu_{k}$. In real-ocean environments, the measurement error is difficult to model because it is influenced by complex environmental factors as well as the relative target position. The general nonlinear function of the measurement is
(11)$$\theta_{k}^{}=g\left(p_{k}^{},\nu_{k}\right),$$

In this paper, the following assumptions were added to the sensor model to focus on the analytic study of the observability requirements for the CoTMA system according to the target–sensor geometric relationship.

The target can be detected in all directions by the sensor (i.e., the censor has no blind spots);The measurement noise $\nu_{k}^{}$ is an additive white gaussian noise (WGN) sequence with a mean of zero and covariance of $R_{k}^{}=\sigma^{2}$ in all directions. The unit σ is a radian;The received target signals propagate directly (i.e., the acoustic ray is not bent);The measurement accuracy cannot be affected by the maneuvering of the sensor (or its towing vehicle).

Although these assumptions are not suitable for calculating sensor trajectory optimization in the real world, that is outside the scope of this paper and will be considered in future work.

Finally, Equation (11) becomes
(12)$$\theta_{k}^{}=h\left(p_{k}^{}\right)+\nu_{k},$$
where $h\left(p_{k}^{}\right)$ is the nonlinear function of $p_{k}^{}$ as
(13)$$h\left(p_{k}^{}\right)=\tan^{-1}\frac{x_{k}}{\sqrt{y_{k}^{2}+z_{k}^{2}}},$$
and the measurement history up until time k is defined as
(14)$$\theta_{}^{k}={\left\{\theta_{i}^{}\right\}}_{i=0}^{k}=\left\{\theta_{0}^{},\theta_{1}^{},\ldots ,\theta_{k}^{}\right\}.$$

## 3. Conventional BoTMA Using the Conic Angle Measurements

In most previous studies of the BoTMA, the state x was represented in the 2D plane rather than in the 3D space. If the $z_{k}^{}$ of Equation (13) is set to 0 for all k, then the CoTMA system from Section 2 can be represented in the 2D plane without approximation and is identical to the 2D BoTMA system. However, if $z_{k}^{}$ is not equal to 0 or is not consistent, it is clear that the CoTMA is different from the 2D BoTMA. This means that a modelling error may occur when the BoTMA system is utilized using the conic angle as measurements.

In this section, the effect of the mismatch between the azimuth and the conic angle is analyzed. An approximation error is defined, and the numerical results of the error are analyzed in terms of a single scan and multiple scans.

### 3.1. Error Analysis for Single Scan

In conventional 2D BoTMA systems, the azimuth in the inertial coordinate system is defined as
(15)$$h^{i}\left(p_{k}^{t,i},p_{k}^{s,i}\right)=\tan^{-1}\frac{x_{k}^{t,i}-x_{k}^{s,i}}{y_{k}^{t,i}-y_{k}^{s,i}}.$$

Additionally, its representation of the body coordinates can be obtained as
(16)$$h_{2D}\left(p_{k}^{}\right)=h^{i}\left(p_{k}^{t,i},p_{k}^{s,i}\right)-\psi_{k}$$
with consideration that $\phi_{k}=\varphi_{k}=0$.

If the conic angle is assumed to be the azimuth to be applied to the BoTMA system in the 2D plane, the approximation error denoted by ε at time k is Equation (16) subtracted from Equation (13) as
(17)$$\varepsilon \left(p_{k}^{}\right)\equiv h_{2D}\left(p_{k}^{}\right)-h\left(p_{k}^{}\right).$$

The approximation error is a function of the relative position in 3D space. In order to visualize the error tendency with respect to the relative position, the parameters $\alpha =y_{k}/x_{k}$ and $\beta =z_{k}/x_{k}$ are defined and can be substituted into Equation (17) as
(18)$$\varepsilon \left(\alpha ,\beta \right)=\tan^{-1}\frac{1}{\alpha }-\tan^{-1}\frac{1}{\sqrt{\alpha^{2}+\beta^{2}}}.$$

Then, the tendency of the approximation error in the αβ domain is represented in Figure 2.

When α is constant, the |ε| is proportional to |β|, which is related to the vertical incidence angle. This means that the error increases as the $x_{k}$ decreases or as the $z_{k}$ increases for the same azimuth. 

The |ε| is inversely proportional to the |α| if β is a nonzero constant. The error decreases if the azimuth of the target becomes closer to the starboard/port direction of the sensor, which is one of the sensor characteristics.

### 3.2. Error Analysis for Multiple Scans

The BoTMA, which uses the conic angle as the azimuth, has two types of the errors. One is the measurement noise $\nu_{k}^{}$ from Equation (12), and the other is the approximation error $\varepsilon \left(p_{k}^{}\right)$ from Equation (17) for every k.

In Section 2, the measurement noise is assumed to be the WGN sequences. Unlike the measurement noise sequence, however, the approximation error is not free of the coloredness because the current target/sensor states depend on their previous states. Furthermore, the average of the approximation error sequence may not be zero in general.

Since most of the estimators assume that the measurement noise is the additive WGN sequences, such approximated measurements that do not match up with the assumption can degrade the estimation accuracy (or tracking performance).

The effect of the approximation error sequence on tracking performance is difficult to interpret analytically. Therefore, it was replaced by numerical analysis by means of a simulation. 

In the simulation scenario, the target and the sensor exist in 3D space, and their trajectories as seen from the horizontal plane are shown in Figure 3a. The initial distance of the target from the sensor on the horizontal plane is 4 km. The target is subjected to the constant velocity motion without the depth changing.

The sensor also maintains its speed in horizontal space, but for the present scenario, the trajectory consists of two legs. In the first leg, the target is located on the port section of the sensor. However, the target is located at the head section of the sensor after the sensor maneuver. It moves to the tail section of the sensor at the end of the scenario.

For the horizontal trajectory scenario, various 3D trajectory scenarios were generated by increasing the value of $z_{k}$, the vertical difference between the target and the sensor, by 10 m from 0 m to 1400 m. In each scenario, $z_{k}$ is a constant for every k. The trajectory scenarios for $z_{k}=0$ m and $z_{k}=1400$ m are depicted on Figure 3b as examples.

The differences between the conic angle and the azimuth histories in $z_{k}=400$ m are shown in Figure 4. The target is located in the head/tail direction of the sensor at the beginning and end of the second leg, respectively, so the difference between the conic angle and the azimuth is conspicuous in the second leg.

The sensor sampling time 20 s, and the measurement noise deviation of the conic angle measurements equals π/180 (=1 deg).

The ML estimator designed for 2D BoTMA is used, and it estimates the initial position and velocity of the target in the horizontal plane. The estimator considers the measurement to be the azimuth, but the measurement is the noise-corrupted conic angle.

As the approximation errors increases, the estimation error will also increase. To obtain statistical estimation errors, a Monte Carlo consisting of 100 trials is used for each scenario.

For the j–th trial in the $z_{k}=d$ scenario, the estimation error of the horizontal plane can be defined as
(19)$$e_{d,j}^{}=\left[\begin{array}{c} {\hat{x}}_{0}^{\left(d,j\right)}-x_{0}^{t,i}\\ {\hat{y}}_{0}^{\left(d,j\right)}-y_{0}^{t,i} \end{array}\right],$$
where ${\hat{x}}_{0}^{\left(d,j\right)}$ and ${\hat{y}}_{0}^{\left(d,j\right)}$ are the estimated elements of the initial target position.

The root mean square (RMS) error for the $z_{k}=d$ scenario can be obtained as
(20)$$e_{d}=\sqrt{\frac{1}{100}\sum_{j=1}^{100}{\left\|e_{d,j}^{}\right\|}^{2}},$$
where ‖A‖ denotes the Euclidean norm of A.

Then, the performance degradation index for the $z_{k}=d$ scenario can be defined as the ratio of the RMS error compared to the RMS error for the $z_{k}=0$ scenario, i.e.,
(21)$$g_{d}=\frac{e_{d}}{e_{0}},$$

If there is no approximation error for the scenario ($z_{k}=0$), then $g_{d}$ is equal to 1.

The tendency of $g_{d}$ is depicted in Figure 5a. For the $z_{k}<500$ interval, $g_{d}$ increases up to 5.2 in proportion to $z_{k}$. In $500\leq z_{k}<800$, $g_{d}$ decreases as $z_{k}$ increases. Although, they are not included in $g_{d}$, errors are still increasing in the bearing and course for this interval. As the influence of the approximation error increases, $g_{d}$ increases exponentially from $z_{k}\geq 800$.

Using Equation (17), the approximation error statistics for each scenario is shown in Figure 5b. It shows that the approximation error is a nonzero mean, and its standard deviation increases as $z_{k}$ increases. Since the statistics for the approximation errors are affected by the relative geometry between the target and the sensor, it is difficult to compensate for the approximation error in advance to utilize the 2D BoTMA algorithms. It is clear that the CoTMA system should be designed in 3D space in order to avoid such approximation errors.

## 4. Observability Analysis for the CoTMA Systems

The CoTMA system has not only the distance ambiguity of a passive sensor but the conic angle ambiguity of a linear array sensor. The conic angle ambiguity is also called “left/right ambiguity” in the horizontal plane.

The system observability includes the possibility of resolving these ambiguities from the given measurements. Therefore, in order to gather meaningful information, an analysis of the system observability should be performed.

This section analyzes the observability of the CoTMA system. The necessary and sufficient conditions to achieve the system observability are derived, and a sensor maneuver strategy is proposed to satisfy the observability requirement.

### 4.1. Fisher Information Matrix (FIM)

The fisher information matrix (FIM) indicates the quantity of information that an observable random variable carries about an unknown parameter. Here, the “observable random variable” is the set of conic angle measurements k, and the “unknown parameter” is the relative target trajectory in the body coordinates, which denotes $x_{}^{k}\equiv {\left\{x_{i}^{}\right\}}_{i=0}^{k}$.

The relative target trajectory can be obtained by using the initial relative state vector in the body coordinates and the given sensor trajectory in the inertial coordinates. Therefore, the “unknown parameter” $x_{}^{k}$ can be summarized as $x_{0}^{}$.

Then, the FIM $J\left(t_{k},t_{0}\right)$ can be defined as
(22)$$J\left(t_{k},t_{0}\right)=-E\left[\frac{\partial^{2}}{\partial x_{0}^{2}}\ln f\left(\theta^{k}|x_{0}^{}\right)|x_{0}^{}\right],$$
where $E\left[\left(\cdot \right)\;|x_{0}^{}\right]$ denotes the conditional expectation of (⋅) for the given $x_{0}^{}$, and $f\left(\theta^{k}|x_{0}^{}\right)$ is the conditional pdf of $\theta^{k}$ for the given $x_{0}^{}$.

Due to statistical independence of the WGN, the conditional pdf $f\left(\theta^{k}|x_{0}^{}\right)$ can be expressed as
(23)$$f\left(\theta^{k}|x_{0}^{}\right)=\prod_{j=0}^{k}f\left(\theta_{j}|x_{0}^{}\right),$$
where $f\left(\theta_{j}|x_{0}^{}\right)=N\left(h\left(p_{j}^{}\right),\sigma^{2}\right)$, and $N\left(m,\sigma^{2}\right)$ denotes the Gaussian pdf with the mean m and covariance $\sigma^{2}$.

In Equation (22), a partial differentiation of the log likelihood pdf of $f\left(\theta^{k}|x_{0}^{}\right)$ exists with respect to $x_{0}^{}$. It is difficult to analytically calculate the partial differentiation using the nonlinear pdf $f\left(\theta^{k}|x_{0}^{}\right)$.

There are various approaches that can be used for the observability analysis of nonlinear systems. Examples of such approaches are the pseudo-linearization method [19,21] and the local-linearization method [22]. Here, the authors use the Jacobian local linearization method, which was utilized in [22].

As such, Equation (22) becomes
(24)$$J\left(t_{k},t_{0}\right)=\sigma^{-2}\sum_{j=0}^{k}{\tilde{H}}_{j}^{T}{\tilde{H}}_{j},$$
where
(25)$$\begin{array}{cc} {\tilde{H}}_{j} & =\frac{\partial^{}}{\partial x_{0}^{}}h\left(x_{j}^{}\right)\\ & =\left[\begin{array}{cccccc} \frac{\partial h\left(x_{j}^{}\right)}{\partial x_{0}^{}} & \frac{\partial h\left(x_{j}^{}\right)}{\partial y_{0}^{}} & \frac{\partial h\left(x_{j}^{}\right)}{\partial z_{0}^{}} & \frac{\partial h\left(x_{j}^{}\right)}{\partial {\dot{x}}_{0}^{}} & \frac{\partial h\left(x_{j}^{}\right)}{\partial {\dot{y}}_{0}^{}} & \frac{\partial h\left(x_{j}^{}\right)}{\partial {\dot{z}}_{0}^{}} \end{array}\right], \end{array}$$
and where $h\left(x_{j}^{}\right)=h\left(p_{j}^{}\right)$ due to $p_{j}^{}\subset x_{j}^{}$.

### 4.2. Observability Analysis with Local Linearization

For the system observability at $t=t_{0}$ from the given information $\theta^{k}$, the FIM $J\left(t_{k},t_{0}\right)$ should be nonsingular for $t_{0}\sim t_{k}$ [21]. The nonsingularity of $J\left(t_{k},t_{0}\right)$ is the same as the columns ${\tilde{H}}_{j}$ having some independence for some $t_{j}$ in $\left[t_{0},t_{k}\right]$.

Then, the observability requirement is
(26)$${\tilde{H}}_{j}\mu \neq 0\;\mathrm{for}\;\mathrm{some}\;t_{0}\leq t_{j}\leq t_{k},$$
where μ is a 6×1 the relative target state vector, which is represented on the body coordinates at $t=t_{0}$ as
(27)$$\mu ={\left[\begin{array}{cccccc} \mu_{px} & \mu_{py} & \mu_{pz} & \mu_{vx} & \mu_{vy} & \mu_{vz} \end{array}\right]}^{T},$$
where the elements of μ are nonzero arbitrary constants.

The Jacobian matrix ${\tilde{H}}_{j}$ can be divided into two partial differential matrices by the chain rule as
(28)$${\tilde{H}}_{j}=\frac{\partial h\left(x_{j}^{}\right)}{\partial x_{j}^{}}\frac{\partial x_{j}^{}}{\partial x_{0}^{}}.$$

The elements of $\frac{\partial h\left(x_{j}^{}\right)}{\partial x_{j}^{}}$ are calculated from Equation (29) through Equation (32)
(29)$$\frac{\partial h\left(x_{j}^{}\right)}{\partial x_{j}^{}}=\frac{\sqrt{y_{j}^{2}+z_{j}^{2}}}{x_{j}^{2}+y_{j}^{2}+z_{j}^{2}}$$
(30)$$\frac{\partial h\left(x_{j}^{}\right)}{\partial y_{j}^{}}=\frac{-x_{j}^{}y_{j}^{}}{\sqrt{y_{j}^{2}+z_{j}^{2}}\left(x_{j}^{2}+y_{j}^{2}+z_{j}^{2}\right)}$$
(31)$$\frac{\partial h\left(x_{j}^{}\right)}{\partial z_{j}^{}}=\frac{-x_{j}^{}z_{j}^{}}{\sqrt{y_{j}^{2}+z_{j}^{2}}\left(x_{j}^{2}+y_{j}^{2}+z_{j}^{2}\right)}$$
(32)$$\frac{\partial h\left(x_{j}^{}\right)}{\partial {\dot{x}}_{j}^{}}=\frac{\partial h\left(x_{j}^{}\right)}{\partial {\dot{y}}_{j}^{}}=\frac{\partial h\left(x_{j}^{}\right)}{\partial {\dot{z}}_{j}^{}}=0.$$

The $\frac{\partial h\left(x_{j}^{}\right)}{\partial x_{j}^{}}$ from Equation (28) is the relative state transition matrix from $t_{0}$ to $t_{j}$. When the sensor moves at a constant velocity, the $\frac{\partial x_{j}^{}}{\partial x_{0}^{}}$ becomes
(33)$$\frac{\partial x_{j}^{}}{\partial x_{0}^{}}=F\left(t_{0,j}\right)=\left[\begin{array}{cc} I_{3} & t_{0,j}I_{3}\\ O_{3} & I_{3} \end{array}\right].$$

Using Equations (29)–(33), the ${\tilde{H}}_{j}\mu$ can be expanded as follows.
(34)$$\begin{array}{cc} {\tilde{H}}_{j}\mu & =\frac{\left(\begin{array}{l} t_{0,j}^{}\mu_{vx}y_{j}^{2}-t_{0,j}^{}\mu_{vy}x_{j}^{}y_{j}^{}-t_{0,j}^{}\mu_{vz}x_{j}^{}z_{j}^{}+t_{0,j}^{}\mu_{vx}z_{j}^{2}\\ +\mu_{px}y_{j}^{2}-\mu_{py}x_{j}^{}y_{j}^{}-\mu_{pz}x_{j}^{}z_{j}^{}+\mu_{px}z_{j}^{2} \end{array}\right)}{{\left(y_{j}^{2}+z_{j}^{2}\right)}^{\frac{3}{2}}}\\ & =\frac{\left\{\begin{array}{l} \left(y_{j}^{2}+z_{j}^{2}\right)\left(\mu_{px}+t_{0,j}^{}\mu_{vx}\right)-x_{j}^{}y_{j}^{}\left(\mu_{py}+t_{0,j}^{}\mu_{vy}\right)\\ -x_{j}^{}z_{j}^{}\left(\mu_{pz}+t_{0,j}^{}\mu_{vz}\right) \end{array}\right\}}{{\left(y_{j}^{2}+z_{j}^{2}\right)}^{\frac{3}{2}}} \end{array}$$

Substituting Equation (13) into Equation (35),
(35)$${\tilde{H}}_{j}\mu =\frac{\mu_{px}+t_{0,j}^{}\mu_{vx}}{\sqrt{y_{j}^{2}+z_{j}^{2}}}-\frac{\tan\;h\left(p_{k}^{}\right)\left\{y_{j}^{}\left(\mu_{py}+t_{0,j}^{}\mu_{vy}\right)+z_{j}^{}\left(\mu_{pz}+t_{0,j}^{}\mu_{vz}\right)\right\}}{y_{j}^{2}+z_{j}^{2}}$$

By substituting Equation (35) into Equation (26) and rearranging it, the observability condition for the CoTMA system is obtained as
(36)$$\tan\;h\left(p_{k}^{}\right)\neq \frac{\left(\mu_{px}+t_{0,j}^{}\mu_{vx}\right)\sqrt{y_{j}^{2}+z_{j}^{2}}}{y_{j}^{}\left(\mu_{py}+t_{0,j}^{}\mu_{vy}\right)+z_{j}^{}\left(\mu_{pz}+t_{0,j}^{}\mu_{vz}\right)}$$
for some $t_{0}\leq t_{j}\leq t_{k}$.

Equation (36) does not hold if the relative target state maintains the constant velocity motion in ${\left[t_{0},t_{j}\right]}^{}$. Therefore, in order to satisfy the observability requirement with the uncooperative target, the sensor maneuver is essential. This result is an extension of the BoTMA observability studies in the 2D plane [18,21].

If Equation (26) is satisfied through the sensor maneuver, it indicates that the estimate of the initial relative target state $x_{0}^{}$ in the body coordinates can be converged using the conic angle measurement history $\theta^{k}$. The degree of the convergence of the estimate is related to the magnitude of the FIM of Equation (24). However, the established system observability does not mean solving the conic angle ambiguity.

Then, the kind of maneuver that is necessary for the sensor to solve the conic angle ambiguity while satisfying Equation (26) must be determined. In the case of a single sensor, course maneuvers should be performed on two or more nonparallel planes. Figure 6 shows an example.

The initial position of the sensor is located at the origin, and its Euler angles are zero. The sensor obtains the conic angle from the target, which moves in constant velocity motion. In the example, two false targets exist, and their trajectories are symmetrical with the target trajectories in the y=0 plane and z=0 plane.

The sensor makes two types of course maneuvers: the first one is a yaw change for 100~190 s, and the second is a pitch change for 300~345 s. The conic angle trajectories with respect to the target and the false targets are shown in Figure 7.

The target and the false targets have the same conic angle trajectory at 0~100 s before the first maneuver of the sensor is performed.

After 100 s, when the change in the yaw angle begins, False Target #1 is separated from the target. This change is the first maneuver on the z=0 plane, but False Target #2, which is symmetrical with the target in the plane, still has the same conic angle trajectory as the target.

In order to distinguish False Target #2 from the target, the sensor needs to maneuver on the plane, which is not parallel to z=0. As the sensor changes the pitch angle after 300 s, the target has a unique conic angle trajectory.

Further numerical simulations are given in Section 6.

### 4.3. Practical Issues

Although the sensor maneuver is needed to satisfy the observability requirements of the CoTMA system, in some applications, the movement of the sensor is either impossible or limited.

For example, although a towed array sonar is able to be moved by a towing boat, in sections where the course is changed, the arrangement shape of the sonar becomes nonlinear, which degrades the measurement accuracy. Additionally, the linear array sensor installed on the seabed for the underwater surveillance system of the harbor is not movable.

In such environments, the system observability requirements can be alleviated by using multiple sensors or by using supported features.

Multiple sensors with different locations can use the intersection between their angles to resolve distance/conic angle ambiguities without requiring the sensor maneuver. This is beyond the scope of this paper.

In special cases, supported features can be obtained, which mitigate the sensor maneuver requirements needed to satisfy the system observability.

If a single sensor is given and a target can be limited to a surface combatant, the diving depth of the sensor is same as the relative depth between the target and the sensor. In this condition, the sensor diving depth can be used as the supporting information needed for target tracking so that the CoTMA system can be expressed in the 2D horizontal plane without the approximation error analyzed in Section 3.

Such an environment is a special but not uncommon CoTMA case, and it is defined as the CDTMA system in this paper.

## 5. Observability Analysis for the CDTMA Systems

Given $z_{k}=d$, where d is a nonzero arbitrary constant, then the conic angle can be calculated by
(37)$$h\left(x_{k}^{},d\right)=\tan^{-1}\frac{x_{k}}{\sqrt{y_{k}^{2}+d_{}^{2}}},$$
where $x_{k}^{}$ is a relative target state vector from the sensor that consists of the position and the velocity in the 2D plane as
(38)$$x_{k}={\left[\begin{array}{cccc} x_{k} & y_{k} & {\dot{x}}_{k} & {\dot{y}}_{k} \end{array}\right]}^{T}.$$

For some $t_{j}$ in $\left[t_{0},t_{k}\right]$, the Jacobian matrix from Equation (37) with respect to Equation (38) for k=0 is
(39)$${\tilde{H}}_{j}=\frac{\partial h\left(x_{j}^{},d\right)}{\partial x_{j}^{}}\frac{\partial x_{j}^{}}{\partial x_{0}^{}},$$
where
(40)$$\frac{\partial^{}h\left(x_{j}^{},d\right)}{\partial x_{j}^{}}=\left[\begin{array}{cccc} \frac{\partial h\left(x_{j}^{},d\right)}{\partial x_{j}^{}} & \frac{\partial h\left(x_{j}^{},d\right)}{\partial y_{j}} & 0 & 0 \end{array}\right],$$
(41)$$\frac{\partial h\left(x_{j}^{},d\right)}{\partial x_{j}^{}}=\frac{\sqrt{y_{j}^{2}+d_{}^{2}}}{x_{j}^{2}+y_{j}^{2}+d_{}^{2}},$$
(42)$$\frac{\partial h\left(x_{j}^{}\right)}{\partial y_{j}^{}}=\frac{-x_{j}^{}y_{j}^{}}{\sqrt{y_{j}^{2}+z_{j}^{2}}\left(x_{j}^{2}+y_{j}^{2}+d_{}^{2}\right)},$$
(43)$$\frac{\partial x_{j}^{}}{\partial x_{0}^{}}=F\left(t_{0,j}\right)=\left[\begin{array}{cc} I_{2} & t_{0,j}I_{2}\\ O_{2} & I_{2} \end{array}\right].$$

Calculating the ${\tilde{H}}_{j}\mu$ of Equation (26) yields
(44)$${\tilde{H}}_{j}\mu =\frac{\left(y_{j}^{2}+d_{}^{2}\right)\left(\mu_{px}+t_{0,j}^{}\mu_{vx}\right)-x_{j}^{}y_{j}^{}\left(\mu_{py}+t_{0,j}^{}\mu_{vy}\right)}{\sqrt{y_{j}^{2}+d_{}^{2}}\left(x_{j}^{2}+y_{j}^{2}+d_{}^{2}\right)}$$

Then, the observability requirement for the CDTMA system is
(45)$$\frac{x_{j}^{}}{\mu_{px}+t_{0,j}^{}\mu_{vx}}\neq \frac{y_{j}^{2}+d_{}^{2}}{y_{j}^{}\left(\mu_{py}+t_{0,j}^{}\mu_{vy}\right)}$$
for some $t_{0}\leq t_{j}\leq t_{k}$.

For d≠0, the Equation (45) is satisfied, even if the sensor performs constant velocity motion. Therefore, the sensor maneuver is not required under the CDTMA system. There is a significant difference between the CDTMA and the BoTMA in the 2D plane.

As mentioned in Section 4.2, however, even if the condition of Equation (26) is satisfied, it cannot be guaranteed to solve the left/right ambiguity. In order to solve the ambiguity without the sensor maneuver, a pair of linear array sensors or information fusion with heterogeneous sensors is still required.

## 6. Numerical Experiments

Simulation I and Simulation II are the numerical observability tests for the CoTMA system determined in Section 4 and the CDTMA system determined in Section 5, respectively. They are focused on the observability analysis tested by the geometric relationship of trajectories between the sensor and the target. The sensors trajectory in this scenario is limited to four typical cases, and the target trajectory is set to the constant velocity motion. For each sensor trajectory, the determinant of the FIM is calculated as the performance index over time.

### 6.1. Simulation I: CoTMA

The initial target position in the inertial coordinates is set to $\left[\begin{array}{ccc} 2000\;m, & 4000\;m & 200\;m \end{array}\right]$, and the target moves at a constant motion velocity of 5 m/s during the scenario. The Euler angles of the target are equal to $\phi_{0}=\varphi_{0}=0$ and $\psi_{0}=-\pi /2$.

The four sensor trajectories for the sensor are considered, and their maneuvering scenarios are list in Table 1. The rest of the sensor parameters with the exception of the maneuvering scenario are the identical. The sensor trajectories begin at the origin of the inertial coordinates with the initial Euler angles of $\phi_{0}=\varphi_{0}=\psi_{0}=0$. The target and the sensor trajectories for 600 s are depicted in Figure 8.

All of the sensors measure the conic angle of the target every 1 s. The standard deviation of the measurement noise σ equals π/180 (=1 deg) for all of the sensors. Since the value of σ only affects the scale of the FIM of Equation (24), it does not have a meaningful effect on the analysis of the system observability. The conic angle histories for all of the sensors are shown in Figure 9.

The performance index of the system observability is the determinant of the FIM ($\left|J\left(t_{k},t_{0}\right)\right|$). If the performance index is equal to zero, the system is not observable due to singularity of the FIM. Otherwise, the inversion of the FIM provides the uncertainty variance for all of the elements to be estimated. The larger the performance index, the smaller the estimated error variance.

Figure 10 shows the performance indices for all of the sensors over time. The left subplot contains the first maneuver interval of “Sensor #3” and “Sensor #4”, while the right subplot includes the other maneuver intervals of “Sensor #4”. The performance index value indicates the amount of information that is needed to estimate the target state.

In the left subplot, the performance index values of “Sensor #3” and “Sensor #4” are nonzero after their maneuvers. At this point, the values are not large yet, but they increase over time. On the other hand, the performance index values of “Sensor #1” and “Sensor #2” are equal to 0. This is because “Sensor #1” and the “Sensor #2” move with a constant velocity (CV) model, and their relative target state vector $x_{k}^{}$ is also changed in the CV model, which cannot satisfy the observability condition of Equation (26). This shows that the sensor maneuver is required to achieve the observability of the CoTMA system.

In the right subplot, the performance index difference between “Sensor #3” and “Sensor #4” increases after the additional maneuvers of “Sensor #4”. At the end of the simulation, the performance index of “Sensor #4” is more than three times larger than that of “Sensor #3”. This result means that the trajectory of “Sensor #4” can obtain relatively more i target tracking information than that of “Sensor #3” can.

Furthermore, “Sensor #4” overcomes the conic angle ambiguity mentioned in Section 4.2 because it maneuvers in two nonparallel planes. However, for “Sensor #3”, its scenario satisfies the system observability but does not solve the cone angle ambiguity. This implies that the estimate for “Sensor #3” can converge on the mirror target trajectory.

### 6.2. Simulation II: CDTMA

The trajectory scenario of the target is the same as in Simulation I, but the sensor trajectories are changed. Two cases are considered: one is a stationary sensor and the other is a sensor that moves in at a CV motion of 5 m/s. The sensors are referred as the “static sensor” and the “constant moving sensor”, respectively, and they originate at the Euler angles of $\phi_{0}=\varphi_{0}=0$ and $\psi_{0}=\pi /3$.

The difference in the vertical depth ($z_{k}$) between the target and the sensors is 200 m, which is given as prior information to the CDTMA system. The simulation takes 600 s; the conic angle of Equation (13) is measured every 1 s, and σ is equal to π/180.

The target and the sensor trajectories are represented in Figure 11, and the conic angle histories with respect to the sensor scenarios are shown in Figure 12.

For each sensor, the performance index of the CoTMA system and that of the CDTMA system are compared. Figure 13 shows performance index tendency for the “static sensor”. As mentioned in Simulation I, the sensor scenario cannot establish the observability in the CoTMA system. However, in the CDTMA system, the performance index value of the trajectory is nonzero, indicating that it satisfies the observability conditions of the CDTMA system.

The tendency of the performance indices for the “constant moving sensor” are shown in Figure 14. The results show that the CDTMA system can achieve system observability without sensor maneuvers.

## 7. Conclusions

In this paper, target localization using conic angle information was defined as the CoTMA system, and the system was modeled in 3D space. The CoTMA system is different from the conventional 2D/3D TMA in the previous studies and is applicable to linear array sonar with a vertical incidence angle.

The observability requirements for the CoTMA system are derived by the local linearized Fisher information matrix, and a sensor maneuver strategy was proposed to achieve system observability. We also proposed the CDTMA system, which can mitigate the sensor maneuver limitation. The results of the analytic scheme were demonstrated through simulation studies.

Based on the results of this paper, the nonlinear tracking algorithm, trajectory optimization, data association, and information fusion will become future topics for CoTMA system research.

## Figures and Tables

**Figure 2 sensors-21-06439-f002:**
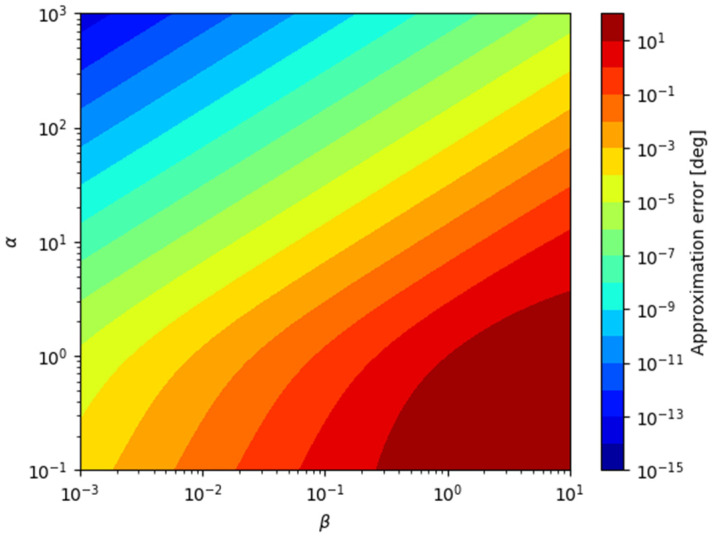
Approximation error ε(α,β) (Log-scale).

**Figure 3 sensors-21-06439-f003:**
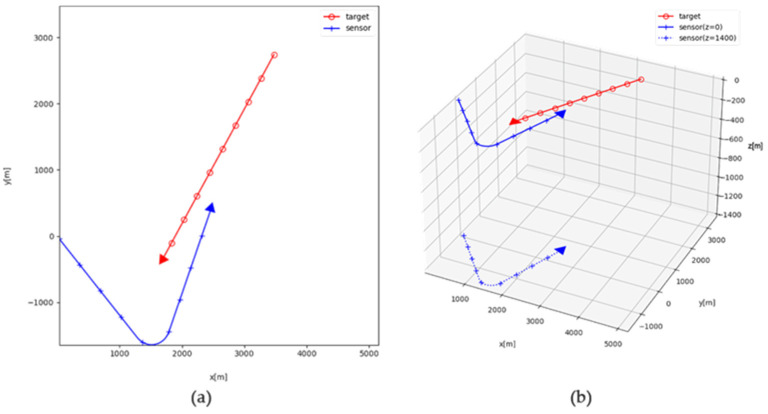
Target and sensor geometry (**a**) in horizontal plane and (**b**) in 3D space.

**Figure 4 sensors-21-06439-f004:**
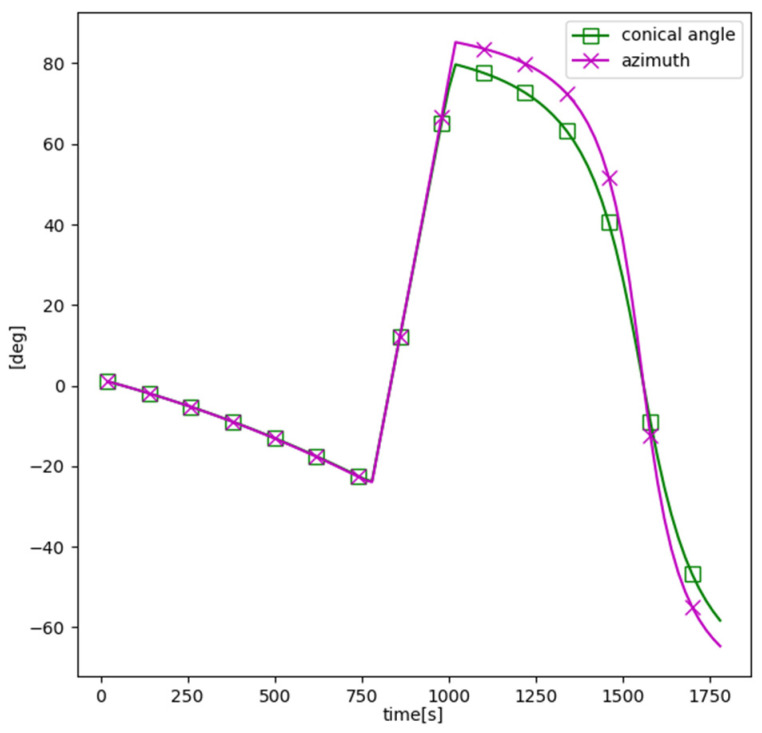
Difference between the conic angle and the azimuth (bearing).

**Figure 5 sensors-21-06439-f005:**
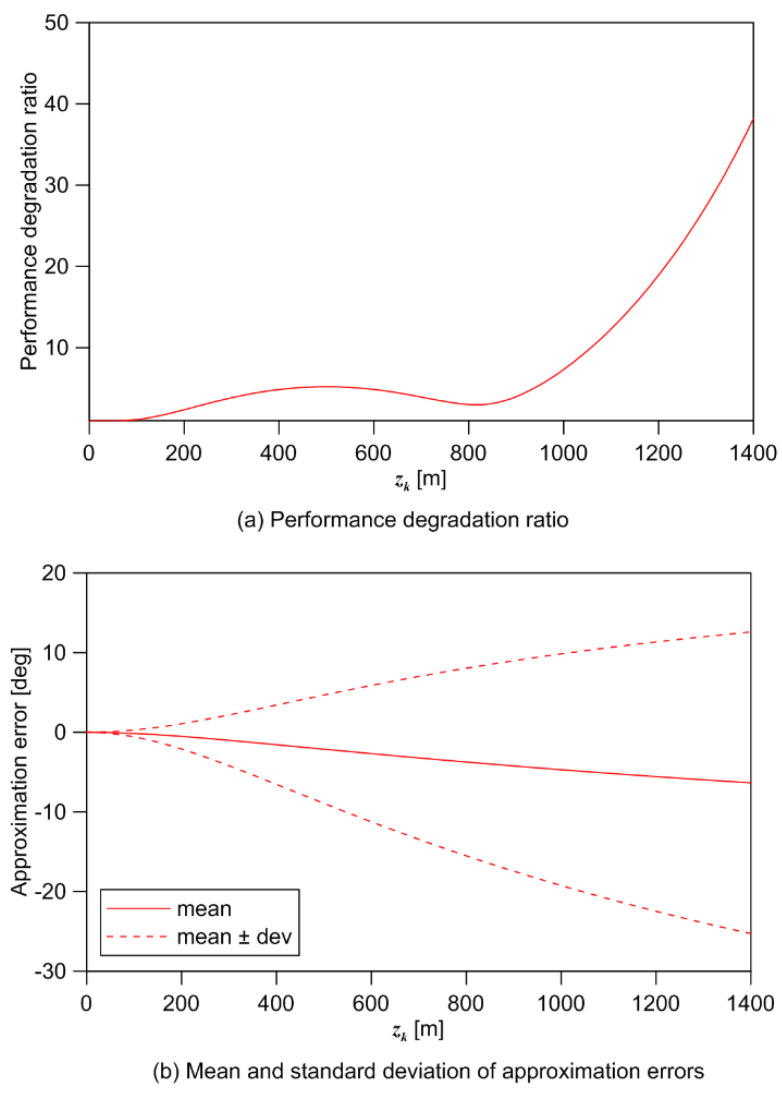
(**a**) Performance degradation ratio and (**b**) the approximation error statistics.

**Figure 6 sensors-21-06439-f006:**
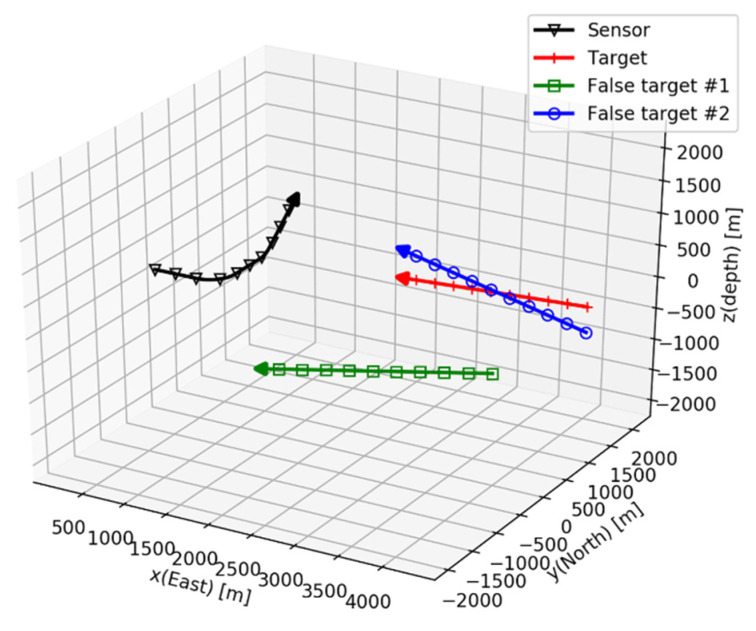
Target and two false targets.

**Figure 7 sensors-21-06439-f007:**
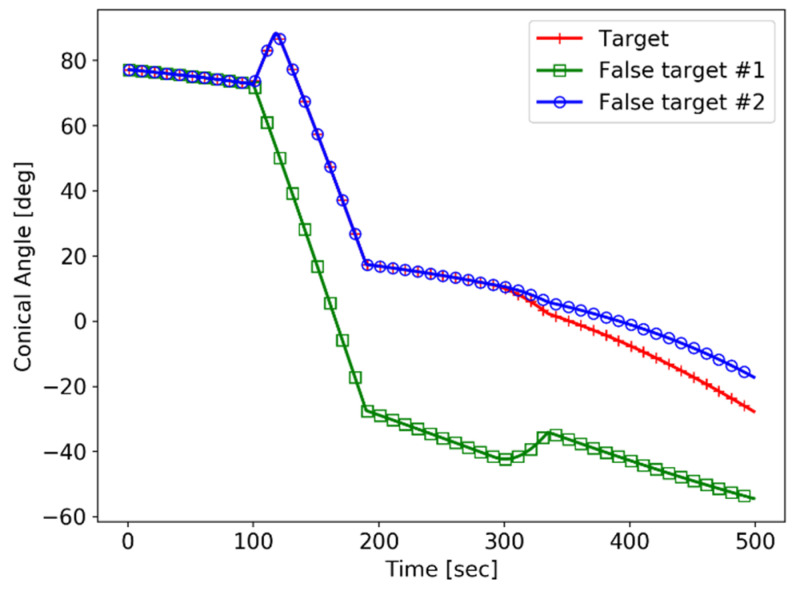
The conic angle trajectories.

**Figure 8 sensors-21-06439-f008:**
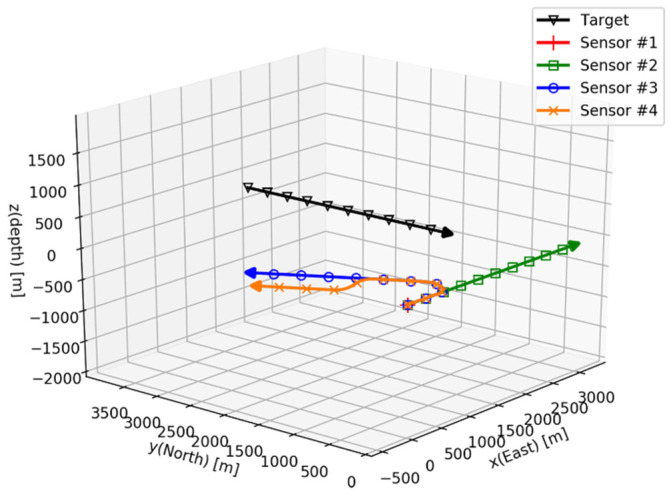
Simulation I: target and sensor trajectories.

**Figure 9 sensors-21-06439-f009:**
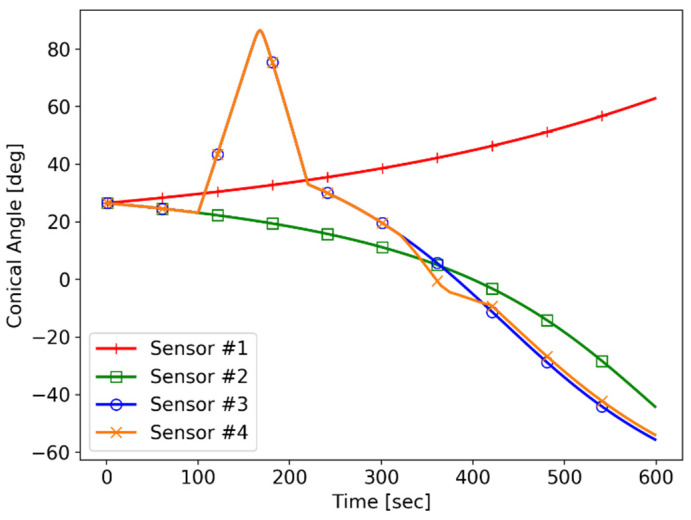
Simulation I: conic angle trajectories.

**Figure 10 sensors-21-06439-f010:**
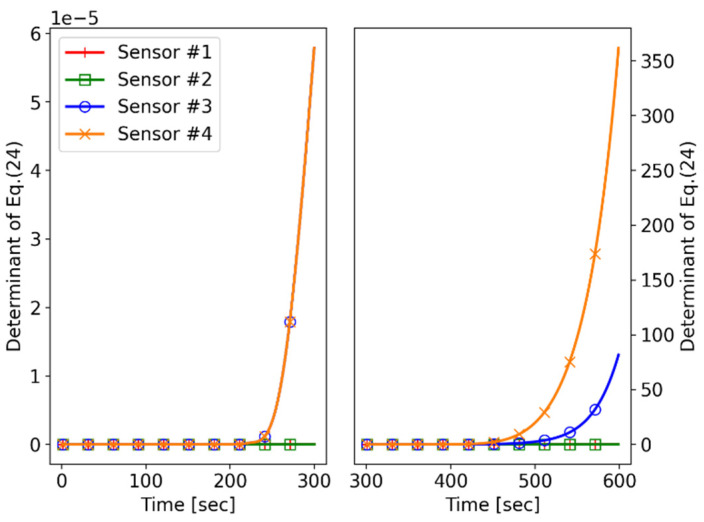
Simulation I: performance index.

**Figure 11 sensors-21-06439-f011:**
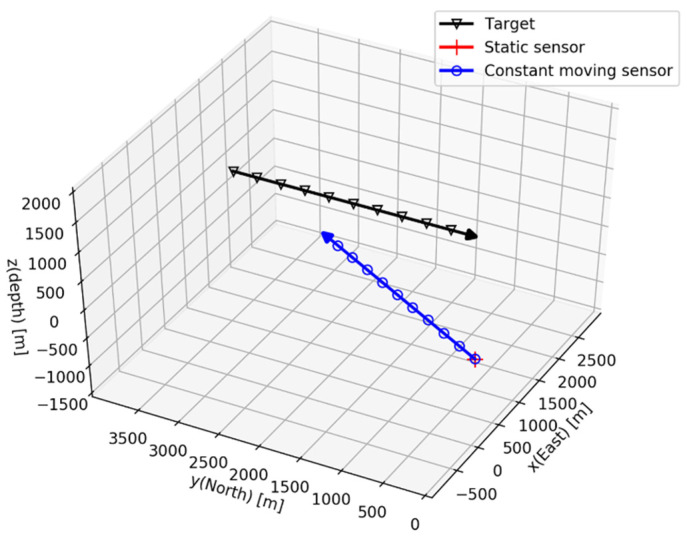
Simulation II: target and sensor trajectories.

**Figure 12 sensors-21-06439-f012:**
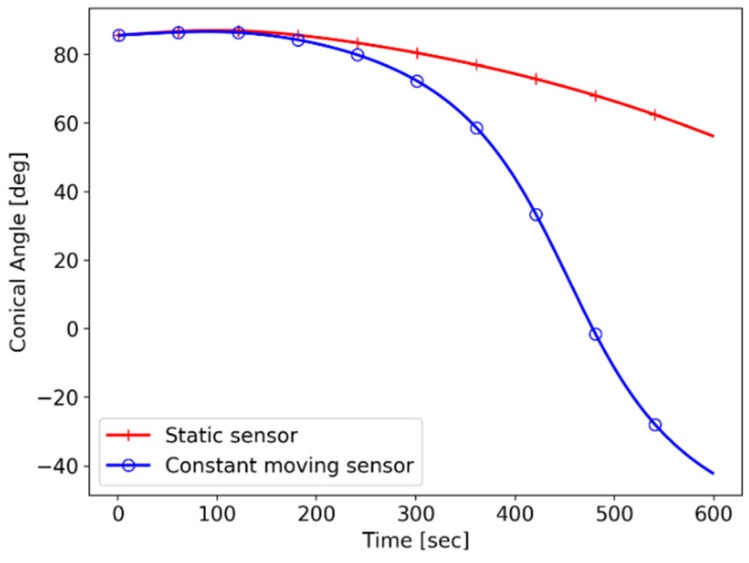
Simulation II: conic angle trajectories.

**Figure 13 sensors-21-06439-f013:**
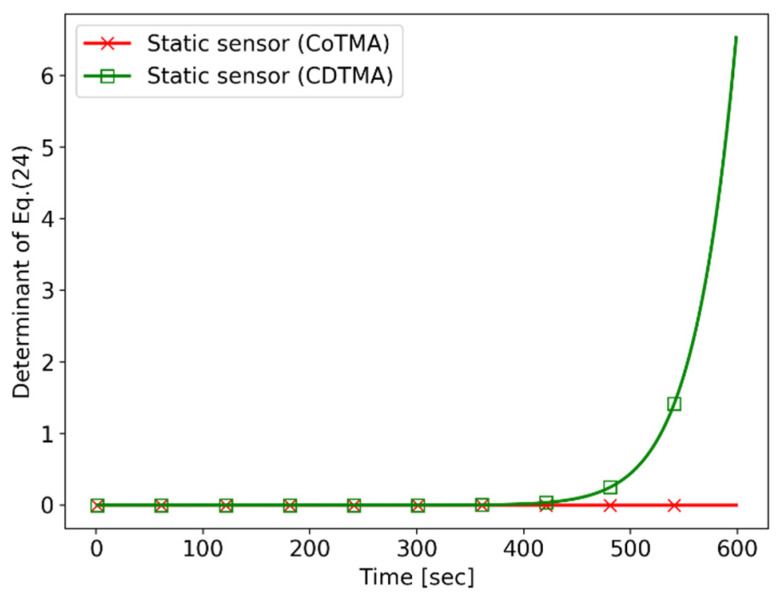
Simulation II: performance index (static sensor scenario).

**Figure 14 sensors-21-06439-f014:**
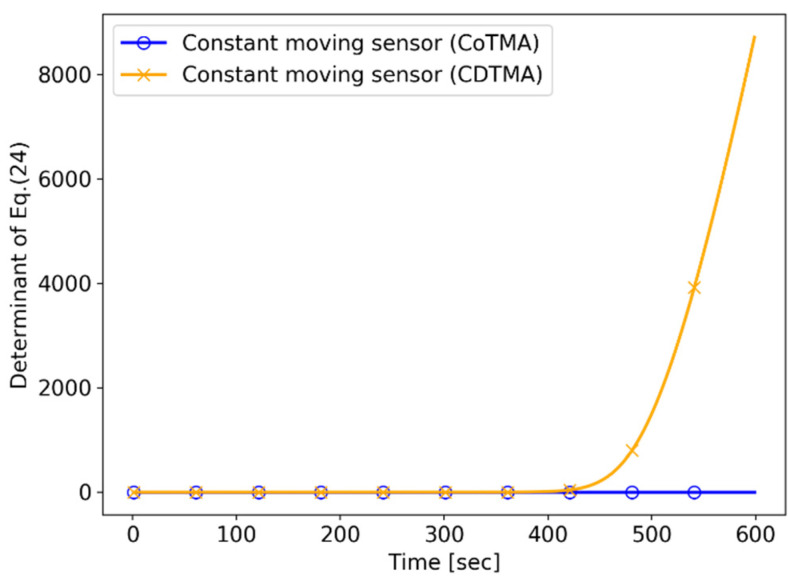
Simulation II: performance index (constant moving sensor scenario).

**Table 1 sensors-21-06439-t001:** Sensor maneuvering scenario.

Time [s]	Sensor #1	Sensor #2	Sensor #3	Sensor #4
0~100	CV ^1^ with 0 [m/s]	CV with 5 [m/s]	CV with 5 [m/s]	CV with 5 [m/s]
100~220			CT ^2^ $\psi :0\to \frac{\pi }{2}$	CT $\psi :0\to \frac{\pi }{2}$
220~320			CV with 5 [m/s]	CV with 5 [m/s]
350~365				CT $\varphi :0\to -\frac{\pi }{4}$
365~375				CV with 5 [m/s]
375~420				CT $\varphi :-\frac{\pi }{4}\to 0$
420~600				CV with 5 [m/s]

^1^ Constant velocity model, ^2^ Coordinated turn model.

## Data Availability

Data sharing not applicable.

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
