# Peer review of "Observability Study on Passive Target Localization by Conic–Angle Measurements"

_sensors, 2021, doi:10.3390/s21196439_

Round 1
Reviewer 1 Report
The content in this paper is divergent so that the true and smart idea could not be given clearly and consistently. My confused understand in the paper is listed as following:
- Compared to BoTMA, would the conic–angle information be used in order to extend only the localization in 2D to 3D?
- There is a big space of derivation in Section 4. However it could be not a major work on CoTMA, because FIM do not appears in Abstract and Conclusion.
- The performance analysis of all numerical experiments is finished according to the determinant of the FIM, would the judge standard be reasonable?
- Some needless descriptions are full in the paper, such as coordinated rotation, measurement error, and sound speed profile.
- The given references are too old, and the newest paper was published in 2009.
- In simulation, CDTMA could mitigate the sensor maneuver limitation. For the known depth information, would CoTMA be used in from 3D to 2D?
- Pay attention to the usage of ‘sensor’, ‘sensors’, and ‘linear array sensor’ in this paper.
Reviewer 2 Report
The authors present observability study of passive target localization with single linear array sensor. For this purpose, observability analysis for 3D conic angles only is considered. The flow of the paper is well in general and easy to follow. However, I have the following concerns for the manuscript:
1. Acronyms are not standardized. Define acronyms where they are first used and following a single style for all acronympes.
2. A comparison with BoTMA from [15] with your work would be interesting.
3. Add a graph for the readers for BoTMA degradation in the estimation process.
4. Since [18] is the baseline for the current study, a comparative analysis with current study is necessary.
5. Show Figure 3 in 3D, so that the trajectories can be viewed properly.
6. Line 201. Is there any difference if the sampling time of the sensor is reduced from 20s.
7. Figure 5. Perfromance degradation ratio is with respect to what exactly?
8. Typos and grammatical errors are found in the manuscript, proofreading and English correction is required.
Reviewer 3 Report
Please see the attached file.

Round 2
Reviewer 1 Report
The author have done a better change. It can been seen in this paper that the author do a solid work, and the proposed method has certain innovative. I agreed to publish.
Author Response
The authors wish to thank the reviewer and the editor for their efforts and contributions in improving the publication.
Reviewer 2 Report
1. By comparison , reviewer means the comparison in the performance of the approach in [15] with the current study.
2. For Figure 5a y-axis, there is not unit.
3. Figure 4 has only caption and no figure
4. Figure 5 contains two captions: one for Figure 4 and other for Figure 5
5. Could authors add a Figure for the impact of sampling time 10s.
Reviewer 3 Report
The reviewer's concerns have been well addressed. It could be accepted in its current version after revising the following minor problem.
The author should check the information of references before submitting the final manuscript:
In [28], "2021 vol. 8, no. 14, pp. 11123-11134." should be revised as "2021, vol. 8, no. 14, pp. 11123-11134."
In [29], "IEEE Wireless Communications, 2021, vol. 10, pp. 251-255." should be revised as "IEEE Wireless Communications Letters, 2021, vol. 10, no. 2, pp. 251-255."
In [30], "IEEE Systems, 2021, vol. 15, no. 2, pp. 2186-2189" should be revised as "IEEE Systems Journal, 2021, vol. 15, no. 2, pp. 2186-2189."
Author Response
Thank you. The information of the above references is corrected in the revised paper.